# SEMI-SUPERVISED LEARNING OF TREE-BASED MODELS USING UNCERTAIN INTERPRETATION OF DATA

## ABSTRACT

Semi-supervised learning (SSL) learns an estimator from labeled and unlabeled data. While diverse methods based on various assumptions have been developed for parametric models, SSL for tree-based models is largely limited to variants of self-training, for which decision trees are not well-suited. We introduce an intrinsic semi-supervised learning algorithm that achieves state-of-the-art performance for tree-based models. The algorithm first grows a tree to minimize a semi-supervised notion of impurity, then assigns leaf values using a leaf similarity graph to optimize either for smoothness or adversarial robustness of the estimator near the data. Our methods can be viewed as natural extensions of conventional tree induction methods emerging from an uncertain interpretation of model input, or alternatively as inductive tree-based approximations of well-established graph-based SSL algorithms.

## 1 INTRODUCTION

In applications of supervised machine learning, there is often copious data available, but labeling the data accurately is expensive. This motivates semi-supervised learning (SSL), which combines strategies from supervised and unsupervised learning to learn a predictive model from labeled and unlabeled data. It does this by leveraging some underlying assumption that informs the use of unlabeled data, such as clusterability, smoothness, or separability of the data. Effective SSL can greatly reduce the amount of labeled data needed to achieve a particular level of performance.

Tree-based models such as decision trees, random forests (Breiman, 2001), ExtraTrees (Geurts et al., 2006), and XGBoost (Chen & Guestrin, 2016) are popular in real-world applications of supervised machine learning for various reasons including ease of use, low resource requirements, very fast inference, reliably good performance on tabular data, and relatively good interpretability. However, most SSL methods are designed for parametric models, so SSL for tree-based models is largely limited to model-agnostic wrapper methods such as self-training (Triguero et al., 2015), for which tree-based models are not even well-suited (van Engelen & Hoos, 2020). This lack of options also limits the choice of assumption that drives the SSL process.

A few SSL methods specific to decision trees have been proposed (Liu et al., 2013; 2015; Levatić et al., 2017); however, these are limited in that their main contribution is choosing splits in a semi-supervised way, and they lack the ability to propagate labels across regions of dense unlabeled data, limiting expressiveness. To fill this gap, we introduce a new SSL algorithm for tree-based models. It first grows a semi-supervised tree as in Levatić et al. (2017), but allows leaves to contain no labeled data, increasing expressiveness since trees may grow large even with very few labeled data. It then assigns leaf values by a novel method: construct a similarity graph over the leaves and use a graph algorithm. We propose two such algorithms corresponding to different underlying assumptions. The first assumes that data with similar feature values should have similar labels, produces a smooth model, solves a linear system to assign leaf values, and is suitable for classification or regression. The second assumes the data is separable into classes by a low-density boundary, produces an adversarially robust model, assigns leaf values using graph min-cut, and is suitable only for classification. Figure 1 motivates the need for this kind of approach even for simple data and highlights the difference between the leaf assignment strategies. The former can be viewed as a tree-based approximation of label propagation (Zhu & Ghahramani, 2002), and the latter is similar in spirit to min-cut SSL approaches (Blum & Chawla, 2001; Blum et al., 2004). These are purely

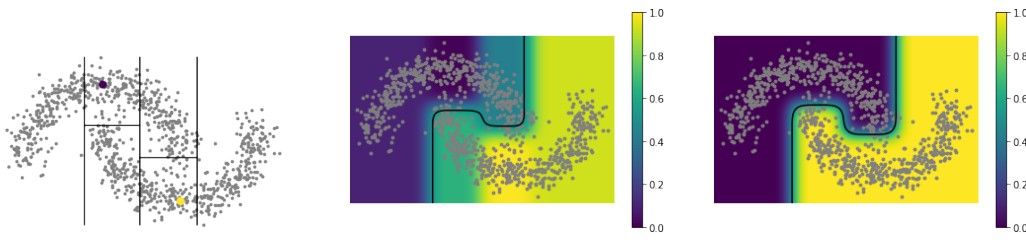

(a) Fit a semi-supervised tree.     (b) Assign smoothed leaf values.     (c) Or assign robust leaf values.

Figure 1: An overview of our SSL algorithm. Gray data are unlabeled. We (a) fit a tree to minimize impurity of both labels and features (Section 3.1), then assign leaf values using one of two strategies with different underlying assumptions, which produce either (b) a smooth model (Section 3.2) or (c) an adversarially robust model (Section 3.3). Previous tree-based SSL methods do not propagate labels to similar leaves, so they cannot learn a good boundary in this example.

graph-based algorithms which are *transductive*, meaning they merely assign labels or pseudo-labels to the unlabeled data without producing a predictive model. Our method, by contrast, is *inductive*, meaning it produces a predictive model. It also scales better computationally with the number of samples, and by controlling tree size, it allows a tradeoff between a tree-like representation with a kind of automatic dimension reduction and a kernel-like representation.

Ultimately, semi-supervised learning algorithms perform largely based on how well their underlying assumption(s) describe the data; by introducing new assumptions for tree-based models—assumptions which are grounded, reasonable, and often used outside tree-based methods—we improve performance on data where other tree-based methods fall short, as shown by our experiment results in Section 4.

## 2 RELATED WORK

Over many years, numerous semi-supervised learning algorithms with various underlying assumptions have been proposed for a wide range of models. We refer the reader to van Engelen & Hoos (2020) for a comprehensive overview. We highlight two taxonomic distinctions: first, a method is *inductive* if it produces a predictive model, or *transductive* if it only assigns labels or pseudo-labels to the unlabeled training data; second, an inductive method is a *wrapper method* if it is agnostic to the predictive model used, or *intrinsic* if it is specialized for a particular class of models. The algorithm presented in this work is thus an intrinsically semi-supervised method for induction of decision trees.

Wrapper methods were once the only option for semi-supervised learning of decision trees and remain a popular approach. Foremost are variants of self-training (Triguero et al., 2015), an iterative process whereby the model is fitted to labeled data, then the most confidently predicted among the unlabeled data are labeled and added to the training pool for the next iteration. Despite its widespread use with trees, tree-based models are actually not well-suited for self-training due to poor calibration and overconfident predictions (van Engelen & Hoos, 2020). As a result, works such as Li & Zhou (2007); Leistner et al. (2009); Deng & Guo (2011); Tanha et al. (2017); Liu et al. (2020) propose strategies to improve self-training of tree-based models.

A few intrinsic methods have been proposed for decision trees. By assigning pseudo-labels to unlabeled data using posterior probability computed from kernel density with reduced dimension on the labeled data, Liu et al. (2013) and Liu et al. (2015) choose better oblique splits using unlabeled data. Similarly, by using impurity of the *features* in addition to impurity of the *labels* as a splitting criterion for tree construction, Levatić et al. (2017) propose semi-supervised predictive clustering trees (SSL-PCTs) that choose better conventional axis-aligned splits using unlabeled data. Our methods leverage SSL-PCTs, but additionally assign leaf values in a semi-supervised way.

Graph-based SSL is a family of transductive methods that constructs a similarity graph over the labeled and unlabeled data and uses a graph algorithm to assign pseudo-labels. We mention two that are relevant to this work. First, label propagation Zhu & Ghahramani (2002) solves a linear system

such that each pseudo-label is a combination of its neighbors' labels or pseudo-labels, weighted by similarity. Second, graph min-cut approaches Blum & Chawla (2001); Blum et al. (2004) use min-cut (equivalently max-flow) algorithms to separate the data into classes such that the inter-class similarity is minimized.

## 3 METHODS

Our proposed semi-supervised learning algorithm has two phases. First, the tree structure is grown using a semi-supervised generalization of CART for KDDTs which is based on the growth strategy of SSL-PCTs from Levatić et al. (2017). Next, a similarity graph over the leaves is constructed and used to assign leaf values. This assignment, differently from previous work, propagates labels across leaves, so all leaves can be assigned a value even if only a few contain labeled data. We propose two assignment approaches; one method implements the assumption that data with similar features should have similar labels, optimizes for smoothness of the model in regions of high density, and conceptually relates to label propagation. The other implements the assumption that the decision boundary should be in low-density regions, optimizes for average adversarial robustness of predictions, that is, distance from the decision boundary, and conceptually relates to graph-based SSL using min-cut. A visual summary is provided in Figure 1.

We notate training data $\boldsymbol{x}_i \in \mathbb{R}^p$, with categorical features one-hot encoded, and labels $\boldsymbol{y}_i \in \mathbb{R}^q$, with each either a vector of $q$ regression targets or a one-hot vector encoding one of $q$ distinct classes. We write $\mathbb{D}_L$ and $\mathbb{D}_U$ the sets of indices of labeled and unlabeled data, respectively, with $\mathbb{D}_L \cup \mathbb{D}_U = \mathbb{D}$ the entire data set with $|\mathbb{D}| = n$.

Underlying our approach is a fuzzy decision tree (FDT) method called Kernel Density Decision Trees (KDDTs) (Good et al., 2022). FDTs allow a decision to take multiple paths with different weight so that a prediction is ultimately a sparse weighted combination of leaf values. KDDTs define the fuzzy splitting by interpreting inputs as uncertain according to a kernel function: interpret each input $\boldsymbol{x}$ as a continuous random variable $\mathbf{x}$ with some probability density $\mathbf{x} \sim f(\cdot, \boldsymbol{x})$. For the KDDT fitting algorithm to work, it is assumed that $f(\cdot, \boldsymbol{x})$ can be written as a product of marginal distributions $f(\boldsymbol{z}, \boldsymbol{x}) = \prod_{j=1}^p f_j(z_j, \boldsymbol{x})$ where $f_j(\cdot, \boldsymbol{x})$ may be different for each $j \in [p]$ and $\boldsymbol{x} \in \mathbb{R}^p$. Typically, $f(\cdot, \boldsymbol{x})$ is a symmetric, unimodal distribution such as multivariate uniform or Gaussian. This $f$ is referred to as a "kernel" for its similarity to the kernel from kernel density estimation. It smooths the model by spreading the input over a local area. The assumptions driving our SSL approaches are formalized by this uncertain interpretation.

### 3.1 SEMI-SUPERVISED TREE GROWTH

Since we do semi-supervised learning under the assumption that data with similar feature values should have similar labels, we grow the tree so that, within each leaf, the labels are pure for labeled data (as in typical supervised tree learning), and the feature values are pure for all data (as in unsupervised tree-based hierarchical clustering). For this purpose we adopt the framework proposed by Levatić et al. (2017) for semi-supervised growth of predictive clustering trees (PCTs), which finds a hierarchical clustering of the data using tree splits. Given a tree $\mathbb{T}$ that partitions $\mathbb{R}^p$ into leaves, each of which is a hyper-interval $\mathbb{L} \subseteq \mathbb{R}^p$, the loss for classification is

$$\frac{1}{n} \sum_{\mathbb{L} \in \mathbb{T}} w_{\mathbb{L}} \left( w \frac{\text{Gini}\{\boldsymbol{y}_i \mid i \in \mathbb{D}_L, \boldsymbol{x}_i \in \mathbb{L}\}}{\text{Gini}\{\boldsymbol{y}_i \mid i \in \mathbb{D}_L\}} + \frac{1-w}{p} \sum_{j \in [p]} \frac{\text{Var}\{x_{i,j} \mid i \in \mathbb{D}, \boldsymbol{x}_i \in \mathbb{L}\}}{\text{Var}\{x_{i,j} \mid i \in \mathbb{D}\}} \right)$$

where $w \in [0, 1]$ is a hyperparameter that controls the tradeoff between supervised and unsupervised loss, with $w = 1$ completely supervised and $w = 0$ completely unsupervised, and $w_{\mathbb{L}}$ is the weight of data belonging to leaf $\mathbb{L}$, defined in the next section. For KDDTs, data can have partial membership in leaves, so the computation of impurity is weighted by membership. For regression, the Gini term is replaced with the average of the variances of the target variables.

Generally we limit tree growth using a cost-complexity pruning (CCP) parameter $\alpha$; if the best split does not result in a loss decrease of at least $\alpha$, then stop growing this subtree and do not perform the split. Since the loss is normalized into $[0, 1]$, a given value of $\alpha$ provides a consistent tradeoff between loss and tree size for various data sets. However, other growth stopping conditions, such as

maximum depth or minimum sample weight in a leaf, are also options. These semi-supervised trees can grow quite large compared to a supervised tree on the same data as a result of the unsupervised component of the impurity. However, once the tree is fitted and the leaf values assigned, it can be pruned according to the purity of the assigned leaf values to get a tree with size in line with fully supervised learning.

## 3.2 SMOOTH LEAF VALUE ASSIGNMENT

As a result of the smoothing induced by the uncertain input interpretation defining KDDTs, a natural SSL approach emerges by simply writing the predictions for unlabeled data and using the resulting system to solve for leaf values. This approach is suitable for both classification and regression. Given a tree $\mathbb{T}$ that partitions $\mathbb{R}^p$ into leaves, a KDDT prediction $\boldsymbol{p}_{\mathbb{T}}(x)$ is derived as follows. Recall that we interpret an input $\boldsymbol{x}$ as random variable $\mathbf{x} \sim f(\cdot, \boldsymbol{x})$.

$$\boldsymbol{p}_{\mathbb{T}}(\boldsymbol{x}) = E_{\mathbf{x}}(\mathbf{y})$$
$$= \int_{\mathbb{R}^p} E(\mathbf{y} \mid \mathbf{x} = \boldsymbol{z}) f(\boldsymbol{z}, \boldsymbol{x}) \, d\boldsymbol{z}$$

In each leaf $\mathbb{L}$, approximate the expected label by the average over the leaf.

$$= \sum_{\mathbb{L} \in \mathbb{T}} \int_{\mathbb{L}} E(\mathbf{y} \mid \mathbf{x} \in \mathbb{L}) f(\boldsymbol{z}, \boldsymbol{x}) \, d\boldsymbol{z}$$

Write membership function $\mu_{\mathbb{L}}(\boldsymbol{x}) = P(\mathbf{x} \in \mathbb{L}) = \int_{\mathbb{L}} f(\boldsymbol{z}, \boldsymbol{x}) \, d\boldsymbol{z}$ and compute the expected label from training data $\boldsymbol{x}_i, \boldsymbol{y}_i, i \in \mathbb{D}$.

$$= \sum_{\mathbb{L} \in \mathbb{T}} \mu_{\mathbb{L}}(\boldsymbol{x}) \frac{\sum_{i \in \mathbb{D}} \boldsymbol{y}_i \mu_{\mathbb{L}}(\boldsymbol{x}_i)}{\sum_{i \in \mathbb{D}} \mu_{\mathbb{L}}(\boldsymbol{x}_i)}$$

Each leaf $\mathbb{L}$ is thus assigned a value $\boldsymbol{v}_{\mathbb{L}} = \frac{1}{w_{\mathbb{L}}} \sum_{i \in \mathbb{D}} \mu_{\mathbb{L}}(\boldsymbol{x}_i) \boldsymbol{y}_i$., where $w_{\mathbb{L}}$ is the total sample weight $w_{\mathbb{L}} = \sum_{i \in \mathbb{D}} \mu_{\mathbb{L}}(\boldsymbol{x}_i)$ at leaf $\mathbb{L}$, and the prediction is $\boldsymbol{p}_{\mathbb{T}}(\boldsymbol{x}) = \sum_{\mathbb{L} \in \mathbb{T}} \mu_{\mathbb{L}}(\boldsymbol{x}) \boldsymbol{v}_{\mathbb{L}}$.

For $i \in \mathbb{D}_U$, $\boldsymbol{y}_i$ is unknown; instead substitute the predicted $\boldsymbol{p}_{\mathbb{T}}(\boldsymbol{x}_i)$.

$$\boldsymbol{v}_{\mathbb{L}} = \frac{1}{w_{\mathbb{L}}} \sum_{i \in \mathbb{D}_L} \mu_{\mathbb{L}}(\boldsymbol{x}_i) \boldsymbol{y}_i + \frac{1}{w_{\mathbb{L}}} \sum_{i \in \mathbb{D}_U} \mu_{\mathbb{L}}(\boldsymbol{x}_i) \boldsymbol{p}_{\mathbb{T}}(\boldsymbol{x}_i)$$
$$= \frac{1}{w_{\mathbb{L}}} \sum_{i \in \mathbb{D}_L} \mu_{\mathbb{L}}(\boldsymbol{x}_i) \boldsymbol{y}_i + \frac{1}{w_{\mathbb{L}}} \sum_{i \in \mathbb{D}_U} \mu_{\mathbb{L}}(\boldsymbol{x}_i) \sum_{\mathbb{L}'} \mu_{\mathbb{L}'}(\boldsymbol{x}_i) \boldsymbol{v}_{\mathbb{L}'}$$
$$= \frac{1}{w_{\mathbb{L}}} \sum_{i \in \mathbb{D}_L} \mu_{\mathbb{L}}(\boldsymbol{x}_i) \boldsymbol{y}_i + \sum_{\mathbb{L}' \in \mathbb{T}} \boldsymbol{v}_{\mathbb{L}'} \frac{1}{w_{\mathbb{L}}} \sum_{i \in U} \mu_{\mathbb{L}}(\boldsymbol{x}_i) \mu_{\mathbb{L}'}(\boldsymbol{x}_i)$$

By stacking this equation over each $\mathbb{L}$, we have a linear system

$$\boldsymbol{V} = \boldsymbol{V}' + \boldsymbol{A}\boldsymbol{V} \tag{1}$$

where $\boldsymbol{V}$ is the unknown matrix of leaf values where (with a slight abuse of notation) each row is the corresponding leaf's value $\boldsymbol{V}_{\mathbb{L},:} = \boldsymbol{v}_{\mathbb{L}}$ for $\mathbb{L} \in \mathbb{T}$, $\boldsymbol{V}'$ is similarly the known matrix of leaf value components from labeled data only $\boldsymbol{V}'_{\mathbb{L},:} = \frac{1}{w_{\mathbb{L}}} \sum_{i \in \mathbb{D}_L} \mu_{\mathbb{L}}(\boldsymbol{x}_i) \boldsymbol{y}_i$ for $\mathbb{L} \in \mathbb{T}$, and $A_{\mathbb{L},\mathbb{L}'} = \frac{1}{w_{\mathbb{L}}} \sum_{i \in \mathbb{D}_U} \mu_{\mathbb{L}}(\boldsymbol{x}_i) \mu_{\mathbb{L}'}(\boldsymbol{x}_i)$ indicates the weight of shared unlabeled samples for each pair of leaves. This can be interpreted as a similarity graph over the leaves. Naturally, it can be solved for $\boldsymbol{V}$ directly, or iteratively as in label propagation.

### 3.2.1 CONVERGENCE AND EQUIVALENCE TO LABEL PROPAGATION

We show that our smoothness-based SSL approach can be interpreted as an approximation of label propagation (Zhu & Ghahramani, 2002) using a kernel that arises from the uncertain input interpretation. Given a kernel $k : \mathbb{R}^p \times \mathbb{R}^p \to \mathbb{R}_{\geq 0}$, label propagation assigns pseudo-labels $\boldsymbol{Y}_U$ by solving the linear system $\boldsymbol{Y}_U = \boldsymbol{K}_{UU}\boldsymbol{Y}_U + \boldsymbol{K}_{UL}\boldsymbol{Y}_L$ where

$$\boldsymbol{K} = \left[ \begin{array}{cc} \boldsymbol{K}_{LL} & \boldsymbol{K}_{LU} \\ \boldsymbol{K}_{UL} & \boldsymbol{K}_{UU} \end{array} \right]$$

is the normalized kernel matrix $K_{i,j} = k(\boldsymbol{x}_i, \boldsymbol{x}_j)/\sum_m k(\boldsymbol{x}_i, \boldsymbol{x}_m)$, split into blocks corresponding to labeled data $\mathbb{D}_L$ and unlabeled data $\mathbb{D}_U$. We write the solution $\boldsymbol{Y}_U(\boldsymbol{K})$ in terms of the kernel similarity matrix $\boldsymbol{K}$.

For given training data $\mathbb{D}$ and tree $\mathbb{T}$, define a kernel

$$k_{\mathbb{D},\mathbb{T}}(\boldsymbol{x}, \boldsymbol{x}') = \sum_{\mathbb{L} \in \mathbb{T}} \frac{\mu_{\mathbb{L}}(\boldsymbol{x})\mu_{\mathbb{L}}(\boldsymbol{x}')}{w_{\mathbb{L}}} = \sum_{\mathbb{L} \in \mathbb{T}} \frac{\int_{\mathbb{L}} f(\boldsymbol{z}, \boldsymbol{x})\, d\boldsymbol{z} \int_{\mathbb{L}} f(\boldsymbol{z}, \boldsymbol{x}')\, d\boldsymbol{z}}{\sum_{i \in \mathbb{D}} \int_{\mathbb{L}} f(\boldsymbol{z}, \boldsymbol{x}_i)\, d\boldsymbol{z}}.$$

and $\boldsymbol{K}_{\mathbb{D},\mathbb{T}}$ the corresponding similarity matrix. Let $\boldsymbol{X}_{U_{i,:}} = \boldsymbol{x}_i$ be the corresponding feature values to labels $\boldsymbol{Y}_U$. Lemma 1 shows that the semi-supervised tree predicts the pseudo-labels from label propagation using kernel $k_{\mathbb{D},\mathbb{T}}$.

**Lemma 1.** $\boldsymbol{Y}_U(\boldsymbol{K}_{\mathbb{D},\mathbb{T}}) = \boldsymbol{p}_{\mathbb{T}}(\boldsymbol{X}_U).$

*Proof.* As shown in its derivation, Equation 1, which is used to solve for leaf values, is equivalent to the system

$$\boldsymbol{p}_{\mathbb{T}}(\boldsymbol{x}_i) = \sum_{\mathbb{L} \in \mathbb{T}} \mu_{\mathbb{L}}(\boldsymbol{x})\boldsymbol{v}_{\mathbb{L}} = \sum_{j \in \mathbb{D}_L} \boldsymbol{y}_j k_{\mathbb{D},\mathbb{T}}(\boldsymbol{x}_i, \boldsymbol{x}_j) + \sum_{j \in \mathbb{D}_U} \boldsymbol{p}_T(x_j) k_{\mathbb{D},\mathbb{T}}(\boldsymbol{x}_i, \boldsymbol{x}_j)$$

for $i \in \mathbb{D}_U$. Noting that $\sum_{j \in \mathbb{D}} k_{\mathbb{D},\mathbb{T}}(\boldsymbol{x}_i, \boldsymbol{x}_j) = 1$ for all $i \in \mathbb{D}$, the above is equivalently written $\boldsymbol{Y}_U = \boldsymbol{K}_{UU}\boldsymbol{Y}_U + \boldsymbol{K}_{UL}\boldsymbol{Y}_L$, where $\boldsymbol{Y}_U = \boldsymbol{p}_{\mathbb{T}}(\boldsymbol{X})$ and $\boldsymbol{K} = \boldsymbol{K}_{\mathbb{D},\mathbb{T}}$. $\qquad\square$

Next we show that these predictions converge as the size of the tree grows large. Define kernel

$$k_{\mathbb{D}}(\boldsymbol{x}, \boldsymbol{x}') = \int_{\mathbb{R}^p} \frac{f(\boldsymbol{z}, \boldsymbol{x})f(\boldsymbol{z}, \boldsymbol{x}')}{\sum_{i \in \mathbb{D}} f(\boldsymbol{z}, \boldsymbol{x}_i)}\, d\boldsymbol{z}$$

with similarity matrix $\boldsymbol{K}_{\mathbb{D}}$. Lemma 2 (proven in Appendix A) shows that, as $\mathbb{T}$ grows large, $\boldsymbol{K}_{\mathbb{D},\mathbb{T}}$ converges to $\boldsymbol{K}_{\mathbb{D}}$.

**Lemma 2.** *If $f$ is continuous almost everywhere and $w > 0$, then $\lim_{|\mathbb{T}| \to \infty} \boldsymbol{K}_{\mathbb{D},\mathbb{T}} = \boldsymbol{K}_{\mathbb{D}}$.*

Finally Theorem 1 concludes that, as the tree grows large, its predictions on the unlabeled data converge to the pseudo-labels assigned by label propagation with kernel $k_{\mathbb{D}}$. In this sense, our proposed SSL algorithm is an inductive tree-based approximation of label propagation. As a result, smaller tree models are more tree-like, while larger tree models are more kernel-like as in label propagation. Controlling tree size enables a tradeoff between the two extremes.

**Theorem 1.** *If $f$ is continuous almost everywhere and $w > 0$, then $\lim_{|\mathbb{T}| \to \infty} \boldsymbol{p}_{\mathbb{T}}(\boldsymbol{X}_U) = \boldsymbol{Y}_U(\boldsymbol{K}_{\mathbb{D}})$.*

*Proof.* By Lemma 1, we have $\lim_{|\mathbb{T}| \to \infty} \boldsymbol{p}_{\mathbb{T}}(\mathbb{D}_U) = \lim_{|\mathbb{T}| \to \infty} \boldsymbol{Y}_U(\boldsymbol{K}_{\mathbb{D},\mathbb{T}})$ with $\boldsymbol{Y}_U$ the solution for label propagation $\boldsymbol{Y}_U(\boldsymbol{K}) = (\boldsymbol{I} - \boldsymbol{K}_{UU})^{-1}\boldsymbol{K}_{UL}\boldsymbol{Y}_L$. Matrix inversion is continuous; therefore, by the properties of limits of sums, products, and compositions, $\lim_{|\mathbb{T}| \to \infty} \boldsymbol{K}_{\mathbb{D},\mathbb{T}} = \boldsymbol{K}_{\mathbb{D}}$ (Lemma 2) implies $\lim_{|\mathbb{T}| \to \infty} \boldsymbol{Y}_U(\boldsymbol{K}_{\mathbb{D},\mathbb{T}}) = \boldsymbol{Y}_U(\boldsymbol{K}_{\mathbb{D}})$. Therefore $\lim_{|\mathbb{T}| \to \infty} \boldsymbol{p}_T(\boldsymbol{X}_U) = \boldsymbol{Y}_U(\boldsymbol{K}_{\mathbb{D}})$. $\qquad\square$

### 3.3 ROBUST LEAF VALUE ASSIGNMENT

Another leaf value assignment strategy, which is only suitable for classification, uses graph min-cut to maximize adversarial robustness, that is, distance from the decision boundary, thereby placing the boundary in low-density regions. While determining the robust radius for a given point is NP-Hard for fuzzy decision trees such as KDDTs (Good et al., 2023), we can use the framework of randomized smoothing to efficiently compute good lower bounds. Randomized smoothing is a technique where a smoothing distribution (usually Gaussian) is used to augment training data with perturbed samples, and predictions are averaged over similar perturbations; KDDTs deterministically achieve the same. Then a lower bound on the robust radius of a prediction is computed as an increasing function of the highest predicted probability. In this sense, the predicted value of a KDDT is directly linked to the robustness of the prediction. For example, for Gaussian smoothing, the robust radius is lower bounded by $r = \sigma\Phi^{-1}(p_{\max})$, where $\Phi^{-1}$ is the inverse Gaussian CDF, $\sigma$ is the standard deviation of the smoother, and $p_{\max}$ is the highest predicted class probability (Cohen et al.,

2019). By maximizing this quantity over training data $\sum_i p_{i,\max}$, we maximize a robustness objective $\sum_i \Phi(r_i/\sigma)$, where each $r_i$ lower bounds the robust radius for sample $i$. Thus we define a semi-supervised objective $\sum_{i\in\mathbb{D}_L} p_{i,y_i} + \sum_{i\in\mathbb{D}_U} p_{i,\max}$ for the model to be *robustly correct* on labeled data and *robust* on unlabeled data. Note that, regardless of practical concerns, it is necessary for the objective to be bounded; if we aim to maximize $\sum_i r_i$ directly, then the ideal model always predicts the majority class and has infinite robust radius for data belonging to that class.

To achieve efficient optimization, we instead maximize $\sum_{i\in\mathbb{D}_L} p_{i,y_i} + \sum_{i\in\mathbb{D}_U} \sum_j p_{i,j}^2$, which lower bounds the original objective and is equal at the extremes $p_{\max} = 1$ and $p_{\max} = p_j \ \forall j$. Before discussing the method of optimization, we first motivate this altered objective from the perspective of minimizing impurity, the typical approach for training tree-based models. For each $i$, let $\mathbf{y}_i \sim \text{Categorical}(\boldsymbol{p}_i)$ be a random label sampled from the predicted label probabilities at $\boldsymbol{x}_i$. The Gini impurity typically used in CART is the probability that a prediction sampled in this way is incorrect: $P_{i\sim\text{Unif}(\mathbb{D}_L)}(\mathbf{y}_i \neq y_i)$. Thus minimizing Gini impurity maximizes the probability that such a prediction is correct. We propose that a natural extension for unlabeled data is to maximize the probability that a prediction on labeled data is correct, and that a prediction on unlabeled data is *consistent*. This targets a high predicted probability, and thus high robustness with KDDTs, for the predicted class on unlabeled data, without prescribing which class should actually be predicted. Let $\mathbf{y}_i$ and $\mathbf{y}_i'$ be independently drawn from $\text{Categorical}(\boldsymbol{p}_i)$.

$$P_{i\sim\text{Unif}(\mathbb{D})}(\mathbf{y}_i = y_i \text{ and } i \in \mathbb{D}_L, \text{ or } \mathbf{y}_i = \mathbf{y}_i' \text{ and } i \in \mathbb{D}_U)$$
$$\propto \sum_{i\in\mathbb{D}_L} p_{i,y_i} + \sum_{i\in\mathbb{D}_U} \boldsymbol{p}_i^T \boldsymbol{p}_i \text{ (the modified robustness objective)}$$
$$= \text{Tr}(\boldsymbol{V'}^T \text{diag}(\boldsymbol{w})\boldsymbol{V} + \boldsymbol{V}^T \boldsymbol{M}^T \boldsymbol{M}\boldsymbol{V})$$

where $\boldsymbol{V'}$ is as in Equation (1), $\text{diag}(\boldsymbol{w})$ is a matrix with vector $\boldsymbol{w} = (w_{\mathbb{L}})_{\mathbb{L}\in\mathbb{T}}$ on the diagonal and 0 elsewhere, and $\boldsymbol{M}$ is the membership matrix $\boldsymbol{M}_{i,j} = \mu_{\mathbb{L}_j}(\boldsymbol{x}_i)$.

With this setup, we can efficiently solve for the leaf values $\boldsymbol{V}$. Since $\boldsymbol{M}^T\boldsymbol{M}$ is positive semidefinite, the objective is convex. Moreover, each row of $\boldsymbol{V}$ must sum to 1 to represent a valid probability distribution over the class labels. This constitutes the maximization of a convex function subject to linear constraints, so at least one maximizer must exist at a corner point; that is, there is some maximizing $\boldsymbol{V}$ with only one nonzero element (which has value 1) in each row.

Let $\boldsymbol{A} = \boldsymbol{M}^T\boldsymbol{M}$. With a slight abuse of notation, let $\boldsymbol{V}_{\mathbb{L},:}$ be the row of $\boldsymbol{V}$ corresponding to leaf $\mathbb{L}$, and similarly for $\boldsymbol{A}$. For each $\mathbb{L}$, let $c_{\mathbb{L}}$ denote the index such that $V^*_{\mathbb{L},c_{\mathbb{L}}} = 1$, that is, $c_{\mathbb{L}}$ is the class predicted at leaf $\mathbb{L}$. Rewrite maximization of the objective as minimization of the following loss.

$$\mathcal{L}(\boldsymbol{V}) = |\mathbb{D}_L| + |\mathbb{D}_U| - \text{Tr}(\boldsymbol{V'}^T \text{diag}(\boldsymbol{w})\boldsymbol{V} + \boldsymbol{V}^T\boldsymbol{A}\boldsymbol{V})$$
$$= \sum_{\mathbb{L}} w_{\mathbb{L}} \sum_c V'_{\mathbb{L},c} + \sum_{\mathbb{L},\mathbb{L}'} A_{\mathbb{L},\mathbb{L}'} - \sum_{\mathbb{L}} w_{\mathbb{L}} V'_{\mathbb{L},c_{\mathbb{L}}} - \sum_{\mathbb{L},\mathbb{L}'} \mathbf{1}\{c_{\mathbb{L}} = c_{\mathbb{L}'}\} A_{\mathbb{L},\mathbb{L}'}$$
$$= \sum_{\mathbb{L},c} \mathbf{1}\{c \neq c_{\mathbb{L}}\} w_{\mathbb{L}} V'_{\mathbb{L},c} + \sum_{\mathbb{L},\mathbb{L}'} \mathbf{1}\{c_{\mathbb{L}} \neq c_{\mathbb{L}'}\} A_{\mathbb{L},\mathbb{L}'}$$

We can construct a graph such that this loss is the value of a $k$-terminal cut, also called a multiway cut. This is a generalization of the min-cut problem where there may be many terminals, whereas standard min-cut has only two—a source and a sink. Given nodes $N = \{s_1, \ldots, s_k, n_1, n_2, \ldots\}$, a $k$-terminal cut is a partition of the nodes into $k$ sets $\mathbb{C}_1, \ldots, \mathbb{C}_k$ such that $s_1 \in \mathbb{C}_1$, $s_2 \in \mathbb{C}_2$, etc. The value of the cut is defined as the total weight of edges removed to separate the nodes into sets: $\sum_{\mathbb{C}\neq\mathbb{C}'} \sum_{n\in\mathbb{C}} \sum_{n'\in\mathbb{C}'} w(n,n')$.

The graph is constructed as follows. For each class $c$, define node $s_c$, and for each leaf $\mathbb{L}$, define node $n_{\mathbb{L}}$. Set edge weights $w(s_c, n_{\mathbb{L}}) = w_{\mathbb{L}} V'_{\mathbb{L},c}$ and $w(n_{\mathbb{L}}, n'_{\mathbb{L}}) = A_{\mathbb{L},\mathbb{L}'}$. Then, given a cut $\mathbb{C}_1, \ldots, \mathbb{C}_k$, for each $\mathbb{L}$, set $V_{\mathbb{L},c_{\mathbb{L}}} = 1$ such that $n_{\mathbb{L}} \in \mathbb{C}_{c_{\mathbb{L}}}$. Then the value of the cut is

$$\sum_{\mathbb{C}\neq\mathbb{C}'} \sum_{n\in\mathbb{C}} \sum_{n'\in\mathbb{C}'} w(n,n') = \sum_{\mathbb{L},c} \mathbf{1}\{c \neq c_{\mathbb{L}}\} w_{\mathbb{L}} V'_{\mathbb{L},c} + \sum_{\mathbb{L},\mathbb{L}'} \mathbf{1}\{c_{\mathbb{L}} \neq c_{\mathbb{L}'}\} A_{\mathbb{L},\mathbb{L}'} = \mathcal{L}(V)$$

and so the minimum such cut provides leaf values $\boldsymbol{V}^*$ that minimize the loss.

For $k = 2$, the problem is simply called *min cut* or *max flow*, and there are many algorithms to solve it with various complexity. The problem of finding the minimum $k$-terminal cut for $k \geq 3$ is NP-Hard; however, a simple heuristic achieves an approximation of $2 - 2/k$ by solving the standard minimum cut problem in a one-vs-rest fashion $k$ times (Dahlhaus et al., 1992). We use this heuristic.

In practice, a model fitted in this way may sacrifice correctness on labeled data for greater robustness on unlabeled data; in the worst case, if there are relatively few labeled samples and the unlabeled data is not easily separable, it is possible that the most robust model just predicts the globally most common label for all inputs. In this case, simply increase the weight of the labeled samples.

## 3.4 MODEL SELECTION

There are three important hyperparameters that must be selected: $w$ controls the amount of supervision in tree growth; the cost-complexity pruning $\alpha$ (or other growth stopping condition) controls the tree size; and the kernel bandwidth $h$ controls the amount of smoothing. If these cannot be chosen by prior knowledge, e.g. setting $h$ according to a desired radius of adversarial robustness for the robust leaf assignment method, then a method for automatic selection is desirable.

Despite the limited number of labeled data in in the semi-supervised setting, we find cross-validation of a supervised metric to the best approach for hyperparameter selection. This is also used by Levatić et al. (2017) to choose $w$. Since the tree growth phase dominates the run time of our methods, we grow the tree just once on all data, then perform $k$-fold cross validation over the labeled part of the data by redoing the leaf assignment phase only. When the number of labeled data is small, accuracy is too coarse, so we use mean absolute error (MAE) instead for model selection. For classification, MAE is $\frac{1}{n} \sum_i 1 - \boldsymbol{y}_i^T \boldsymbol{p}(\boldsymbol{x}_i)$.

## 3.5 ENSEMBLES

Tree-based models are often ensembled to improve generalization to unseen data, and while KDDTs can generalize better than standard decision trees, they also benefit from ensemble approaches (Good et al., 2022). For some ensemble methods, the extension to semi-supervised learning is straightforward. Random forests, perhaps the most popular tree ensemble, use bagging, where each model is trained on a bootstrap sample of the data, and at each split, only a random subsample of features is considered. We follow the precedent of Levatić et al. (2017) and adapt this to semi-supervised learning by bootstrap sampling from the union of the labeled and unlabeled data, then train the semi-supervised trees on each bootstrap sample with feature subsampling. Another popular tree ensemble algorithm, ExtraTrees (Geurts et al., 2006), only changes the tree fitting process by selecting thresholds at random, and can be used for SSL without modification. Boosted tree ensembles, however, including algorithms such as AdaBoost (Freund & Schapire, 1997) and XGBoost (Chen & Guestrin, 2016), use a supervised loss function, so the adaptation to SSL is not so straightforward. We leave this topic to future work.

## 3.6 COMPUTATIONAL COST

In practice, the run time is dominated by the tree growth phase. This semi-supervised tree growth suffers from adverse complexity with respect to the number of features $p$; each split iterates over each feature, and for each feature, impurity must be computed for each feature and class or regression target, resulting in $O(np(p + q))$ to find an optimal split.

We use the following strategies to mitigate this cost. First, control the tree size by avoiding very small CCP-$\alpha$; also, for efficient tuning of $\alpha$, grow the tree just once using the smallest $\alpha$, then prune it to get the model for larger $\alpha$. Second, when training random forests, sample just $\log_2 p$ features to consider for splitting instead of the more commonly used $\sqrt{p}$. Normally when fitting a random forest, if a suitable split is not found in these $\log_2 p$ features, the search continues over the other features, and growth stops only if none of the $p$ features yield an acceptable split. Thus the cost of such a search added over the leaves is still $O(np(p + q))$. We instead stop after just the initial $\log_2 p$ features regardless of whether an acceptable split was found, reducing the cost to $O(n(p + q) \log p)$.

Some additional cost is incurred during tree growth and inference because, in KDDTs, a sample may belong to more than one decision path. For a kernel with unbounded support, every sample belongs

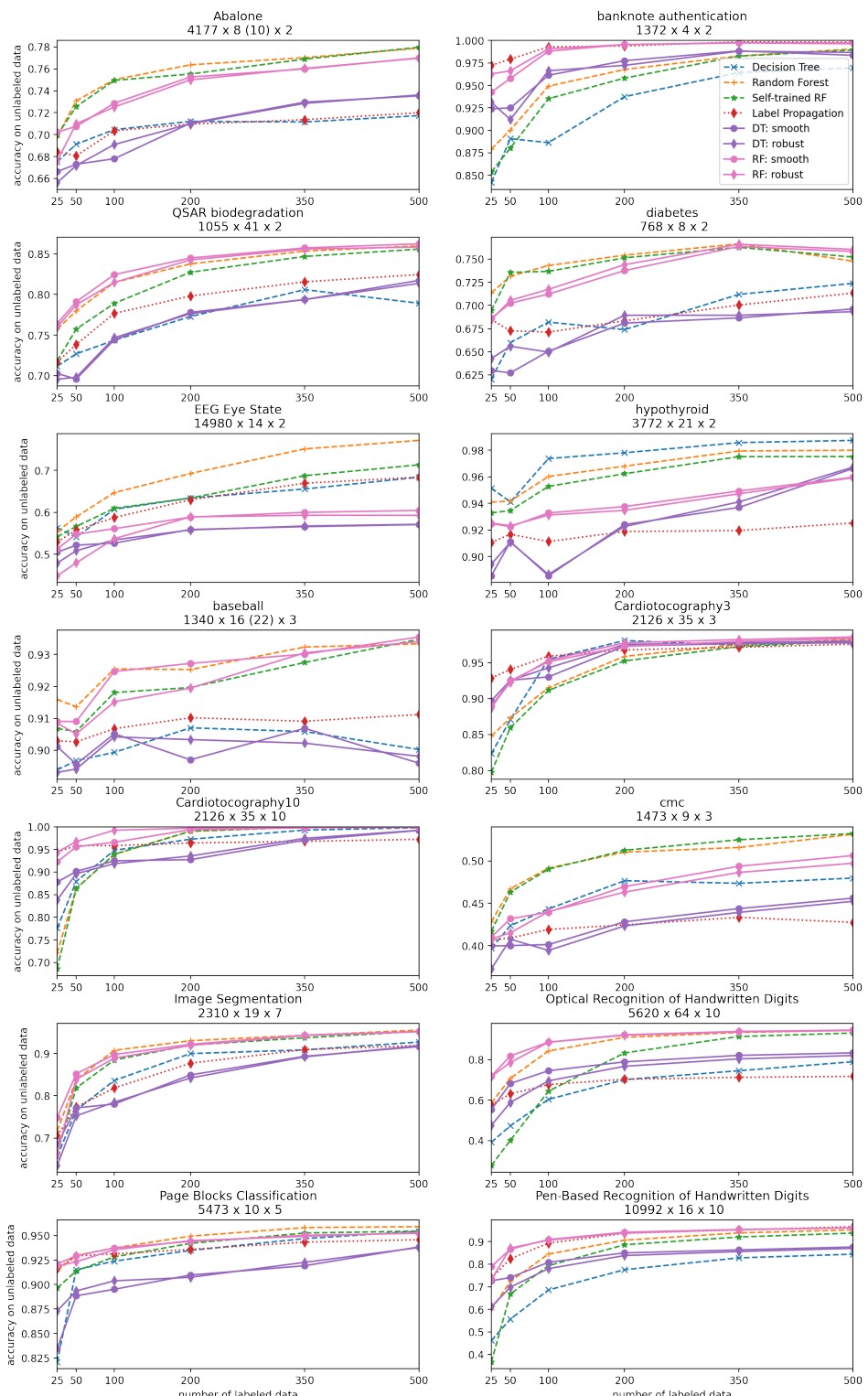

Figure 2: Average accuracy on unlabeled data vs. number of labeled data. Solid lines show our models. Beneath each data set name is shown its number of instances, number of features (after one-hot encoding categorical features, if applicable), and number of classes.

to every leaf, so growth and inference time scale linearly with the number of nodes; however, with a reasonably scaled kernel of bounded support, membership in leaves is sparse, so cost scales more in line with a crisp decision tree, that is, linearly with the depth for a balanced tree.

For leaf value assignment, the complexity is the same as the related graph-based methods, namely label propagation and min-cut SSL, but the size of the graph is equal to the number of leaves instead of the number of data. Therefore, for fixed tree size, our methods' total cost scales linearly with the number of data, better than any purely graph-based SSL. These results suggest that our methods may fill a practical niche wherein there are very many samples but not too many features.

## 4 EXPERIMENTS

We replicate the experimental setup of Levatić et al. (2017). Of their data sets, we use a subset containing those that (1) we could find available online, (2) are not primarily categorical features, and (3) have few enough features that our methods execute in reasonable time. For each data set, we train our models and several baselines with various numbers of data chosen uniformly at random to be labeled. We report average accuracy of predictions on the unlabeled data used in training over 5 different samples of labeled data. The baselines include regular supervised decision tree and random forest, self-trained random forest, and label propagation. Hyperparameter and data preprocessing details are in Appendix B. While we are not able to include results from the SSL-PCTs of Levatić et al. (2017), the matching experimental setup at least enables some comparison.

Figure 2 shows the results. To summarize, among tree-based methods, one of ours outperforms the baselines for at least one label count on 11 of 14 data sets, and outperforms them in all label counts on 4 of 14 data sets. In aggregate, one of our models is the best tree-based model in 50% of cases. Smoothed random forest has the best average rank at 2.47, followed by random forest at 2.52 and robust random forest at 3.02. In cases where our methods perform poorly, we see that label propagation also performs poorly; this is unsurprising since they rely on similar assumptions. In most cases, random forests greatly outperform single-tree models. There is usually little difference in performance between our two leaf value assignment strategies. Neither is preferred for every data set, but for a given data set, one tends to be consistently better than the other if there is a gap.

To keep the experiment run times down, we use CCP-$\alpha$ no smaller than $10^{-3}$ for our models, which substantially limits tree size. This and the modifications to the random forest feature sampling described in Section 3.6 reduce the power of our models, which we suspect negatively influences performance on complex data. While this limitation was necessary for comprehensive benchmarking, it is unlikely to be necessary in practice except perhaps for data with many features.

## 5 CONCLUSIONS

We have proposed a novel intrinsic method for semi-supervised learning of decision trees. By propagating label information across similar leaves, it improves expressiveness and implements different assumptions from previous tree-based SSL methods and, as a result, performs well on many data sets where those methods fall short. It is similar to graph-based semi-supervised learning relying on Euclidean distance; however, unlike graph-based approaches, it produces a predictive model that has all the practical benefits of decision trees, scales well with data volume, and ultimately outperforms graph-based assignment of labels on several data sets.

The key remaining limitation is the scalability of the proposed method vs. the dimensionality of data. While this can be partially mitigated using dimension reduction and strategies such as those described in Section 3.6, our future work will focus on improving the efficiency of fitting unsupervised and semi-supervised PCTs, which would have impact beyond just this work. Future work will also investigate the proposed approach for regression tasks and consider its adaptation to other tree-based models, such as boosted ensembles.

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

## A    PROOF OF LEMMA 2

Since $f(\cdot, \boldsymbol{x})$ is continuous almost everywhere, it is uniformly continuous almost everywhere on a compact domain, So we have for every $\epsilon > 0$, there exists $\delta > 0$ such that, for any leaf $\mathbb{L}$ of diameter (maximum range of its constituent intervals) less than $\delta$ on which $f(\cdot, \boldsymbol{x})$ is continuous, we have $\max_{\boldsymbol{z} \in \mathbb{L}} f(\boldsymbol{z}, \boldsymbol{x}) - \min_{\boldsymbol{z} \in \mathbb{L}} f(\boldsymbol{z}, \boldsymbol{x}) < \epsilon$. Likewise for $\boldsymbol{z} \mapsto f(\boldsymbol{z}, \boldsymbol{x}) f(\boldsymbol{z}, \boldsymbol{x}') / \sum_{i \in \mathbb{D}} f(\boldsymbol{z}, \boldsymbol{x}_i)$. Let $\boldsymbol{c} \in \mathbb{L}$. Then, if we consider $\mathbb{T}$ to partition a compact domain $\mathbb{X} \subseteq \mathbb{R}^p$,

$$
\begin{aligned}
\frac{\int_{\mathbb{L}} f(\boldsymbol{z}, \boldsymbol{x}) \, d\boldsymbol{z} \int_{\mathbb{L}} f(\boldsymbol{z}, \boldsymbol{x}') \, d\boldsymbol{z}}{\sum_{i \in \mathbb{D}} \int_{\mathbb{L}} f(\boldsymbol{z}, \boldsymbol{x}_i) \, d\boldsymbol{z}} &= \frac{\text{mean}_{\boldsymbol{z} \in \mathbb{L}} f(\boldsymbol{z}, \boldsymbol{x}) \text{Vol}(\mathbb{L}) \text{mean}_{\boldsymbol{z} \in \mathbb{L}} f(\boldsymbol{z}, \boldsymbol{x}') \text{Vol}(\mathbb{L})}{\sum_{i \in \mathbb{D}} \text{mean}_{\boldsymbol{z} \in \mathbb{L}} f(\boldsymbol{z}, \boldsymbol{x}_i) \text{Vol}(\mathbb{L})} \\
&= \frac{\text{mean}_{\boldsymbol{z} \in \mathbb{L}} f(\boldsymbol{z}, \boldsymbol{x}) \text{mean}_{\boldsymbol{z} \in \mathbb{L}} f(\boldsymbol{z}, \boldsymbol{x}')}{\sum_{i \in \mathbb{D}} \text{mean}_{\boldsymbol{z} \in \mathbb{L}} f(\boldsymbol{z}, \boldsymbol{x}_i)} \text{Vol}(\mathbb{L}) \\
&= \frac{(f(\boldsymbol{c}, \boldsymbol{x}) + O(\epsilon))(f(\boldsymbol{c}, \boldsymbol{x}') + O(\epsilon))}{\sum_{i \in \mathbb{D}} (f(\boldsymbol{c}, \boldsymbol{x}_i) + O(\epsilon))} \text{Vol}(\mathbb{L}) \\
&= \frac{f(\boldsymbol{c}, \boldsymbol{x}) f(\boldsymbol{c}, \boldsymbol{x}') + O(\epsilon)}{\left( \sum_{i \in \mathbb{D}} f(\boldsymbol{c}, \boldsymbol{x}_i) \right) (1 + O(\epsilon))} \text{Vol}(\mathbb{L}) \\
&= \frac{f(\boldsymbol{c}, \boldsymbol{x}) f(\boldsymbol{c}, \boldsymbol{x}') + O(\epsilon)}{\sum_{i \in \mathbb{D}} f(\boldsymbol{c}, \boldsymbol{x}_i)} (1 + O(\epsilon) + O(\epsilon)^2 + \dots) \text{Vol}(\mathbb{L}) \\
&= \frac{f(\boldsymbol{c}, \boldsymbol{x}) f(\boldsymbol{c}, \boldsymbol{x}')}{\sum_{i \in \mathbb{D}} f(\boldsymbol{c}, \boldsymbol{x}_i)} \text{Vol}(\mathbb{L}) + O(\epsilon \text{Vol}(\mathbb{L}))
\end{aligned}
$$

and likewise

$$\int_{\mathbb{L}} \frac{f(\boldsymbol{z}, \boldsymbol{x}) f(\boldsymbol{z}, \boldsymbol{x}')}{\sum_{i \in \mathbb{D}} f(\boldsymbol{z}, \boldsymbol{x}_i)} \, d\boldsymbol{z} = \text{mean}_{\boldsymbol{z} \in \mathbb{L}} \frac{f(\boldsymbol{z}, \boldsymbol{x}) f(\boldsymbol{z}, \boldsymbol{x}')}{\sum_{i \in \mathbb{D}} f(\boldsymbol{z}, \boldsymbol{x}_i)} \text{Vol}(\mathbb{L})$$

$$= \left( \frac{f(\boldsymbol{c}, \boldsymbol{x}) f(\boldsymbol{c}, \boldsymbol{x}')}{\sum_{i \in \mathbb{D}} f(\boldsymbol{c}, \boldsymbol{x}_i)} + O(\epsilon) \right) \text{Vol}(\mathbb{L})$$

$$= \frac{f(\boldsymbol{c}, \boldsymbol{x}) f(\boldsymbol{c}, \boldsymbol{x}')}{\sum_{i \in \mathbb{D}} f(\boldsymbol{c}, \boldsymbol{x}_i)} \text{Vol}(\mathbb{L}) + O(\epsilon \text{Vol}(\mathbb{L}))$$

so we have error

$$|k_{\mathbb{D}, \mathbb{T}}(\boldsymbol{x}, \boldsymbol{x}') - k_{\mathbb{D}}(\boldsymbol{x}, \boldsymbol{x}')| = \sum_{\mathbb{L} \in \mathbb{T}} O(\epsilon \text{Vol}(\mathbb{L}))$$

$$= O(\epsilon \text{Vol}(\mathbb{X})).$$

Since $f$ is continuous almost everywhere, finitely many leaves contain discontinuities, so their error goes to zero as $\delta \to 0$. So, for any compact $\mathbb{X}$, we can achieve bounded error with leaves of diameter less than $\delta$. Then as $\mathbb{X} \to \infty$ and $\delta \to 0$, we have $k_{\mathbb{D}, \mathbb{T}}(\boldsymbol{x}, \boldsymbol{x}') \to k_{\mathbb{D}}(\boldsymbol{x}, \boldsymbol{x}')$. Thus it is sufficient to show that leaf diameter goes to zero. An exception to this requirement is that, for any $\mathbb{L}$ where $f_j(z_j, \boldsymbol{x}_i) = f(z_j', \boldsymbol{x}_i)$ for all $\boldsymbol{z}, \boldsymbol{z}' \in \mathbb{L}$, $i \in \mathbb{D}$, the error is zero over feature $j$, so the leaf need not be any smaller in dimension $j$. And, in fact, the tree growth algorithm will not split on feature $j$ in such cases because no further gain is possible.

Otherwise, consider a feature $j$ where these conditions are not met, and where splitting results in some gain. Then there must be some sufficiently small $\epsilon$ such that, if the range of each $f_k(\cdot, \boldsymbol{x}_i)$, $k \neq j$, $i \in \mathbb{D}$ is less than $\epsilon$, then the gain from splitting on feature $k$ is less than the gain from splitting on feature $j$. Since $f_k(\cdot, \boldsymbol{x}_i)$ is continuous almost everywhere, there exists a $\delta$ such that this is true with a diameter less than $\delta$ for all $k \neq j$; then the next split is on feature $j$. In this way, the leaf diameters along all necessary dimensions approach zero as the size of the tree grows large.

## B    EXPERIMENT DETAILS

Here we provide details of experimental setup for reproducibility.

The data is preprocessed by one-hot encoding categorical variables, replacing missing values (there are not many in these data sets) with the mean, and standardizing all features to mean 0 and standard deviation 1. When we sample data to be labeled in the training of models, we ensure that at least one instance of each class is represented, that is, we sample one instance uniformly at random from each class, then the rest uniformly at random (without replacement) from the entire remaining data set.

At the time of writing, each data set is available from the UCI Machine Learning Repository (Kelly et al.) or OpenML (Vanschoren et al., 2013) at the following URLs:

- Abalone
  https://archive.ics.uci.edu/dataset/1/abalone

- banknote authentication
  https://archive.ics.uci.edu/dataset/267/banknote+authentication

- QSA biodegradation
  https://archive.ics.uci.edu/dataset/254/qsar+biodegradation

- diabetes
  https://www.openml.org/d/37

- EEG Eye State
  https://archive.ics.uci.edu/dataset/264/eeg+eye+state

- hypothyroid
  https://www.openml.org/d/57

- baseball
  https://www.openml.org/d/185

- Cardiotocography
  https://archive.ics.uci.edu/dataset/193/cardiotocography

- cmc
  https://www.openml.org/d/45056

- Image Segmentation
  https://archive.ics.uci.edu/dataset/50/image+segmentation

- MiceProtein
  https://www.openml.org/d/40966

- Optical Recognition of Handwritten Digits
  https://archive.ics.uci.edu/dataset/80/optical+recognition+of+handwritten+digits

- Page Blocks Classification
  https://archive.ics.uci.edu/dataset/78/page+blocks+classification

- Pen-Based Recognition of Handwritten Digits
  https://archive.ics.uci.edu/dataset/81/pen+based+recognition+of+handwritten+digits

Our baseline models are implementations from the popular machine learning package scikit-learn Pedregosa et al. (2011). Hyperparameters are as follows.

- **Decision tree**. We use `sklearn.tree.DecisionTreeClassifier` with default hyperparameters. The tree is fully grown.

- **Random forest**. We use `sklearn.tree.DecisionTreeClassifier` with default hyperparameters. Each has 100 fully grown trees, and features to consider for splitting are randomly subsampled to $\sqrt{p}$ at each split. If a suitable split is not found, the remaining features are searched.

- **Self-training Random Forest**. We use `sklearn.tree.DecisionTreeClassifier` as above along with `sklearn.semi_supervised.SelfTrainingClassifier` with default hyperparameters.

- **Label propagation**. We use `sklearn.semi_supervised.LabelPropagation` using a RBF (Gaussian) kernel with standard deviation chosen from $[0.01, 0.0215, .0464, 0.1]$. We choose the standard deviation that minimizes the loss defined in the original label propagation paper Zhu & Ghahramani (2002).

- **Our methods**. Our KDDTs use a Gaussian kernel. For fitting, a piecewise-constant kernel is required, so we use a histogram approximation of Gaussian with 7 pieces truncated to 3 standard deviations. We choose CPP-$\alpha$ from $[0.001, 0.00316, 0.01, 0.0316, 0.1]$ and kernel bandwidth $h$ (standard deviation of the Gaussian kernel) from $[0.01, 0.0215, .0464, 0.1]$ using 10-fold cross validation of the mean absolute error (MAE) on the unlabeled data. This range for CCP-$\alpha$ limits trees to be relatively small, which is necessitated by the run time and scope of the experiments; it is a current weakness of this analysis. For efficient cross validation, a we grow a single tree structure on all data and only reassign the leaf values for each fold. For the supervised vs. unsupervised tradeoff parameter $w$, we simply use $w = |\mathbb{D}_L|/|\mathbb{D}|$. This way it is fully unsupervised with zero labeled data and fully supervised with zero unlabeled data. During leaf assignment, we increase the weight of labeled data to $\max(1, |\mathbb{D}_U|/|\mathbb{D}_L|)$. This gives at least as much total weight to labeled data as unlabeled data to prevent collapse to the global majority if the data is not easily separable.

