# OpenReview forum: "Semi-Supervised Learning of Tree-Based Models Using Uncertain Interpretation of Data"
_ICLR.cc/2024/Conference — ICLR 2024 Conference Withdrawn Submission_

### Official Review · Reviewer_ExDS · 2023-10-19

**Soundness:** 3 good
**Presentation:** 3 good
**Contribution:** 2 fair
**Rating:** 3
**Confidence:** 3

**Summary:**

This paper introduces a semi-supervised learning (SSL) algorithm for tree-based models. The proposed algorithm employs an intrinsic SSL approach that optimizes the estimator by leveraging the inherent structure of tree-based models. Specifically, it constructs a leaf similarity graph to capture similarities between leaf nodes of the tree and uses this graph to propagate label information from labeled to unlabeled data.

**Strengths:**

The paper is readable and well-written. The insight that the smoothness-based SSL approach can be seen as an approximation of label propagation using a kernel derived from the uncertain input interpretation is interesting. It provides a higher level of clarity regarding how the proposed method approximates label propagation using trees.

**Weaknesses:**

The proposed method combines various techniques, and it's unclear how beneficial each individual contribution is. While incorporating various strategies can be practically beneficial, I believe that in academic papers, there is a need for micro-level analyses of each component.

Furthermore, since the algorithm is built with assumptions about the data, it performs well on data that aligns with those assumptions and poorly on data that does not. While this is evident, and the performance was often better on the datasets used in this experiment, there remain questions about the significance of this study. I believe that if there were advantages such as computational cost or support from theoretical superiority, they could serve as strong points beyond performance.

Only prediction accuracy has been measured, and there has been no comparison in terms of computation time or memory usage. Although the order has been evaluated, a comparison with other methods is necessary.
Upon reviewing the Appendix, it appears that the proposed method conducts parameter tuning and other fine-grained optimizations with great precision compared to other methods. I'm curious about the time required for these parameter search processes in comparison to other methods. It would be helpful to understand the impact of both the algorithm itself and the thorough parameter tuning on the results.

(Minor comment: The abbreviation 'KDDT' is used in the first paragraph before its definition is provided in the third paragraph of Section 3.)

**Questions:**

(See weakness part)

1: Could you provide ablation study results? In addition, there are various degrees of freedom when conducting experiments with the proposed method, such as the design of kernels in KDDT or the design of histograms when considering piecewise constant kernels. How would the experimental results change with variations in these designs?

2: Could you report the cost required for parameter search processes in comparison to other methods?

---

> ### Author Response · Authors · 2023-11-15
> **Response to review**
>
> In response to weaknesses:
>
> We are now including as baselines fully supervised KDDTs and semi-supervised PCTs, which are the contributing components that can be used on their own. Thanks for this suggestion.
>
> We believe that improved performance, sometimes by a large margin, is a strong case for the value of the work. Beyond that, we do have reduced computational complexity in the number of samples compared to purely graph-based methods, and we implement similar assumptions while producing a predictive model which is tree-based and has additional utility as a result. We are not sure how to compare fundamentally different methods from a theoretical perspective, as you say, but we are open to suggestions. We do at least thoroughly motivate our approach from a theoretical standpoint.
>
> A consistent measurement of training time is difficult because we are running experiments in parallel on shared resources with variable hardware. Our method is the most time consuming, due largely to the increased cost of fitting with a variance objective and partially also due to using fuzzy trees and selecting a kernel bandwidth hyperparameter. Tree-based methods in general have low memory needs, and ours should not be different, so we don't see a need for a deep analysis of this. Could you clarify the statement: "Although the order has been evaluated, a comparison with other methods is necessary." We tune two hyperparameters, and the search is not fine-grained; it is actually quite coarse. Tuning over the cost-complexity parameter alpha is not at all expensive because it is done by pruning, which avoids repeating the most expensive component of our algorithm, which is fitting trees. As for kernel bandwidth, we use only four candidate values. The other methods do not require hyperparameter choices; this is simultaneously a relative weakness of our method, in that there is some cost of hyperparameter tuning, but also a strength in that these hyperparameters, particularly the kernel bandwidth, adjust the semi-supervised behavior in meaningful ways that are not possible in some of the baseline models, which is perhaps a reason we achieve improved performance. We will note this in the revision.
>
> In response to questions:
>
> Yes, we will give ablation results in the revision by including baselines for each component of the approach. We do not see a need to do ablation, however, for the kernel shape and training kernel approximation. In general, we always use a Gaussian kernel, and the way it is made into a histogram for training is less influential and is ultimately just a tradeoff of cost and approximation error, so we don't think there is a reason to run additional experiments for this. The choice of kernel bandwidth and ccp-alpha are important, however. Would a breakdown of performance across these hyperparameter values address your concern?
>
> There is no hyperparameter search for the current baselines, though there will be for some of the baselines we are adding in the revision. As for our methods, the main cost of tuning is that it necessitates cross-validation, which we make less expensive by doing it only on the leaf assignment phase, which is much less time consuming than tree fitting. If abundant computation resources are available, one of course could and should do cross-validation by refitting the model entirely. As for each parameter, the cost of tuning alpha is negligible since it can be done by pruning the tree; the cost of tuning the bandwidth roughly corresponds to the number of bandwidths we try. In our case, we try 4 bandwidth values, to the cost is roughly 4x. We will note these nuances in the revision.

---

### Official Review · Reviewer_YpyQ · 2023-10-21

**Soundness:** 3 good
**Presentation:** 2 fair
**Contribution:** 2 fair
**Rating:** 5
**Confidence:** 4

**Summary:**

The paper addresses the problem of training decision trees in a semi-supervised setting, i.e. only a fraction of the data are associated with labels. The authors root their proposal in label propagation, i.e. the classifier being trained is used to replace the missing labels. Two strategies for label propagation are considered: the first one (smooth leaf value assignment) amounts to replace the missing labels with the probabilistic predictions of the tree, while the second one makes use of randomized smoothing and seemingly consists in maximizing consistency between the missing label replacements and available labels.

The paper features discussions on how the tree can be regularized, on how the SSL tree algorithm can be plugged into a random forest, and on the computational complexity of the approach. The proposals are then succinctly evaluated and compared on several datasets of the literature, before a conclusion is made.

**Strengths:**

The paper addresses an interesting problem, since training a tree in a semi-supervised setting remains open.

The proposals are technically good, i.e. the math are correct.

**Weaknesses:**

The paper is not very well written. It lacks a clear presentation of the existing works upon which it builds (and/or a clear distinction between these works and the proposal); the proposals lack formalization; some mathematical objects are not properly introduced or described. Overall, it seems to lack structure.

The paper focuses on the technical side rather than on the interpretation. As such, it well addresses how to solve the optimization problems presented (as I already stated above), but not (or not always) why these problems in particular were derived.

Last, I have a mixed feeling towards the experiments. Indeed, the paper addresses training trees in a semi-supervised setting, the aim of which is to learn a classifier able to generalize well on new data. However, the accuracy of the model is evaluated on the unlabeled training data, which were used to train the model. This seems highly criticizable to me.

**Questions:**

The writing and the presentation of the paper should be improved. Some acronyms are not defined (e.g., KDDT), some mathematical objects are not properly introduced (e.g., \bf{y}). There are a few typos or clumsy expressions ("there is often copious data available", "we interpret [...] as random variable"), and several sentences are hard to understand ("By maximizing this quantity over training data $\sum_i p_{i,\text{max}}$, we maximize a robustness objective [...]", or "smaller tree models are more tree-like, while larger tree models are more kernel-like"). Some references (e.g., the first and the last) are incomplete.

The description of the existing works is quite succinct, and may have been made more precise—and more formal. As well, some parts of the proposal (such as, notably, Section 3.3) lacks formalization, which does not help understanding nor fully appreciating its interest.

Some notations are not appropriate (such as, e.g., the probability density attached to an instance, i.e. $f(\cdot,\boldsymbol{x})$)—note that this interpretation of an instance as the realization of a continuous random variable $\bf{x}$ may have been discussed further. It would have been nice to better present the matrices at the end of Section 3.2 (right before Sec. 3.2.1).

Lemma 1 is not clear; it should be readable on its own. The first propagation strategy being equivalent to label propagation as per Zhu and Ghahramani, what is the added value ?

To strategies were proposed to "identify" missing labels: how do they compare to each other ? An experimental comparison, if not a theoretical study, would have been interesting.

There seems to be a major problem in the evaluation of the method. In the experiments, it seems that accuracy is computed over the unlabeled training data, not on some test data. This does not seem rigorous, as the actual generalization capacity is not evaluated, and the estimate thus obtained is biased since these data are used in training.

---

> ### Author Response · Authors · 2023-11-15
> **Response to review**
>
> In response to weaknesses:
>
> In what way can we improve the presentation of existing work? We have referenced, summarized, and described the relationship of our work to other works with related methods. Besides some notational corrections and clarifications, we believe our presentation of the proposed approach is sufficiently formalized - are there specific concerns we can address? We also have a clear sectional structure describing the components of the method. Please let us know if there was confusion, and how we can make it clearer.
>
> Within each section, we motivate the proposed method; in particular, the leaf assignment methods are based on clear assumptions of smoothness and separability, respectively, which are commonly used in SSL methods for other model types. Please let us know if there are specific elements that should be further clarified.
>
> We gave transductive evaluation based on the precedent of the closest related method, Levatic et al. (2017), and because it makes it possible to compare to label propagation. However, based on multiple reviewers' requests, we will instead report inductive performance in the revision and, accordingly, leave out label propagation as a baseline.
>
> In response to questions:
>
> We will clarify the specific items you pointed out. Thanks for this feedback. Please let us know if there are other ways we can improve clarity.
>
> We will give more thorough descriptions of related work, but due to the page limit, it may need to be moved to the appendix. We will also strive to improve section 3.3. We are unsure what you mean by a lack of formality in the related works and in section 3.3. What could help make this more formal?
>
> We see nothing wrong with the notation mentioned - it is common to represent probability density as a function f, and it is the notation used in the original KDDT paper. To clarify, KDDTs do not interpret the inputs as the realization of a random variable; they interpret each input as a random variable itself. We write the distribution of the random variable associated with an input $x$ as $f(\cdot,x)$; in our experiments, it is the Gaussian probability density with center x and standard deviation chosen by cross-validation.
>
> Would making the lemma statements self-contained address this concern? compared to label propagation, the added value is that, unlike label propagation, our method is (1) inductive, (2) tree-based, (3) based on a kernel that is data-adaptive and not tractable to compute fully for use with label propagation, (4) does not require computation of pairwise distance between data points, and (5) is more performant on many data sets in practice, as shown in the experiments. We will clarify these points in the revision.
>
> We gave transductive evaluation based on the precedent of the closest related method, Levatic et al. (2017), and because it makes it possible to compare to label propagation. However, based on multiple reviewers' requests, we will instead report inductive performance in the revision and, accordingly, leave out label propagation as a baseline.

---

> > ### Comment · Reviewer_YpyQ · 2023-11-21
> > **Follow-up**
> >
> > I'd like to thank the authors for their answers to my comments.

---

### Official Review · Reviewer_e7qU · 2023-10-30

**Soundness:** 3 good
**Presentation:** 2 fair
**Contribution:** 2 fair
**Rating:** 3
**Confidence:** 5

**Summary:**

The paper introduces a method for learning trees in semi-supervised learning setting. This is done in two steps: 1) greedily grow a tree with modified gini splitting criterion; 2) use similarity-based graph to assign labels for each leaf. The authors propose two approaches for the second step: 1) smooth label assignment that relies on solving a linear system; 2) an assignment based on graph min-cut.

The method is applicable to both classification and regression trees, and easily extendable to ensembles. Experiments demonstrate superior performance compared to baselines.

**Strengths:**

- the paper is well-motivated. In the era of data-driven decision making, we observe extensively increasing data with limited supervision. Thus, the paper considers a practical setup.
- the method is applicable to both classification and regression trees, and easily extendable to ensembles.
- Experiments demonstrate superior performance compared to baselines.

**Weaknesses:**

Major comments:
- **Novelty**. An important and pertinent paper [1] appears to be absent from the list of references. It is worth noting that the tree growing procedure outlined in this work closely follows the methodology established by Levatic et al. However, the principal innovation lies in the leaf assignment process. The first (out of 2) technique involves solving a linear system on a similarity matrix, akin to the principles of graph Laplacian-based methods used in [1]. The authors demonstrate its equivalence to label propagation techniques. Notably, [1] explicitly builds upon this methodology, and it is noteworthy that one iteration in [1] bears a striking resemblance to the first approach elucidated in this paper. This correlation underlines the significance of [1] in the context of the current study.
- **Experiments**. As noted above, [1] is highly related to the proposed approach and should be compared as a baselines. Moreover, used benchmarks a somewhat limited in their sizes and do not resemble real-world semi-supervise scenario.
- **Scalability**. [1] employs a sparse graph representation, necessitating the utilization of sparse linear algebra techniques for solving the linear system. However, it is important to acknowledge that in this particular context, scalability could potentially pose a challenge, primarily attributable to the dense representation of matrix A. This consideration prompts a thoughtful examination of the trade-offs associated with the chosen approach, especially in cases where computational resources may be constrained.


[1] Zharmagambetov, A. and Carreira-Perpinan, M. A. (2022): "Semi-supervised learning with decision trees: Graph Laplacian tree alternating optimization". Advances in Neural Information Processing Systems 35 (NeurIPS 2022), pp. 2392-2405.

**Questions:**

- The authors mention that the second approach is more robust against adversarial attacks. Just wondering whether it was demonstrated experimentally as I was unable to find?

---

> ### Author Response · Authors · 2023-11-15
> **Response to review**
>
> In response to weaknesses:
>
> - We were not aware of this paper - thanks for pointing it out. We will include it in the discussion of related work. We would point out that our method does *not* construct a similarity graph over the data, but over the leaves; we never compute distance between any two data instances. Also, the tree-based simplification of the implied similarity score, as described in Lemma 1, is one of the reasons we think our method can perform better than a conventional graph-based approach. We will point out these differences in the revision.
> - As far as I can tell, there is no code given with the referenced paper, and the method does not look straightforward to implement. Moreover, it appears to depend on parametric learning (of decision trees), which seems like a whole different model class from what we are doing here. Please correct me if I am wrong.
> - We have acknowledged computational limitations in the paper, mainly with respect to the tree growth phase, which is the dominant computational cost. We could use sparse graph methods, but the speedup would be insubstantial since this step is not the dominant computational cost. In what ways could we better clarify the scaling properties of our approach?
>
> In response to questions:
>
> - It was not demonstrated experimentally. Measuring adversarial robustness exactly is NP-Hard for tree ensembles [1] and fuzzy decision trees [2], so measuring it experimentally may be difficult, and beyond that, no verification method is known for KDDTs. However, at least for single-tree KDDT-based models, we can provide a lower bound on robust radius using randomized smoothing theory. We will consider what kind of empirical comparison we can make given these limitations and add it to the revision.
>
> [1] Kantchelian, Alex, J. Doug Tygar, and Anthony Joseph. "Evasion and hardening of tree ensemble classifiers." In International conference on machine learning, pp. 2387-2396. PMLR, 2016.
>
> [2] Good, Jack H., Nicholas Gisolfi, Kyle Miller, and Artur Dubrawski. "Verification of Fuzzy Decision Trees." IEEE Transactions on Software Engineering (2023).

---

> > ### Comment · Reviewer_e7qU · 2023-11-20
> >
> > > We would point out that our method does not construct a similarity graph over the data, but over the leaves; we never compute distance between any two data instances. Also, the tree-based simplification of the implied similarity score, as described in Lemma 1, is one of the reasons we think our method can perform better than a conventional graph-based approach. We will point out these differences in the revision.
> >
> > Thanks for pointing out the differences. However, I still believe that these two approaches are quite similar, with the given approach calculating the similarity score between tree leaves rather than between data points and tree induction procedure closely following Levatic et al.
> >
> > > As far as I can tell, there is no code given with the referenced paper, and the method does not look straightforward to implement.
> >
> > True, however they evaluate the performance on traditional and well-known benchmarks (e.g. MNIST). So, it would be straightforward to make apple-to-apple comparison using similar setup.
> >
> > > Moreover, it appears to depend on parametric learning (of decision trees), which seems like a whole different model class from what we are doing here. Please correct me if I am wrong.
> >
> > Based on my understanding, the parametric formulation is simply the method by which they devise the learning algorithm (e.g., using TAO). However, during inference, it will still be a traditional decision tree.
> >
> > > In what ways could we better clarify the scaling properties of our approach?
> >
> > One way is to demonstrate by empirical evidences, i.e. evaluate and report runtime metrics on somewhat larger benchmarks...
> >
> > --------------
> >
> > The authors mention a revision in the general response. Are you referring to future plans? I was unable to find a revised version...

---

> > > ### Author Response · Authors · 2023-11-21
> > > **Thanks**
> > >
> > > Thanks for the additional feedback.
> > >
> > > We had planned a revised version, but we have decided instead to target a future venue so that we have enough time to fully incorporate all the feedback from these reviews.

---

### Official Review · Reviewer_HSUS · 2023-11-01

**Soundness:** 2 fair
**Presentation:** 4 excellent
**Contribution:** 2 fair
**Rating:** 3
**Confidence:** 3

**Summary:**

The submission presents two methods for semi-supervised learning of decision trees, based on the tree construction method of predictive clustering trees, which have been applied to semi-supervised learning in earlier work, and the inference method from kernel density decision trees. The proposed two approaches apply graph-based semi-supervised learning strategies, label propagation and min-cut, and enable semi-supervised learning to propagate class information across similar leaves. Experiments evaluate the two methods for varying amounts of labeled training data on 14 datasets, for both stand-alone trees and random forests, and compare to supervised decision trees and random forests, self-trained random forests, and graph-based label propagation with an RBF kernel. The results are quite mixed when comparing the proposed methods to the purely supervised competitors, and there is no clear winner.

**Strengths:**

The submission is nicely written.

There is a lack of work on intrinsic methods for semi-supervised learning of decision trees, which is addressed by this work.

Theoretical results are proven for one of the two methods that are presented.

**Weaknesses:**

The experimental results do not show a consistent advantage compared to simple supervised baseline decision trees and random forests respectively. Moreover, this is in spite of the extensive hyperparameter tuning using cross-validation for the proposed methods and the absence of kernel smoothing in the decision tree baselines. The proposed approaches have two hyperparameters that are tuned using cross-validation.

There is a lack of an ablation study where the proposed method is run in a purely supervised mode (for both stand-alone decision trees and random forests).

There is no comparison to basic predictive clustering trees for semi-supervised learning.

No significance tests are performed and no confidence intervals are provided.

The proposed methods are advertised as inductive ones, but the evaluation appears to be transductive.

Cross-validation is applied for parameter tuning based on a tree structure learned from the entire dataset. This will yield overfitting. Computational complexity arguments are not very strong because the cross-validation can be trivially parallelized.

The submission references several papers that attempt to improve self-training in trees (e.g., Lie et al., 2020), but the experiments only include the basic variant of self-trained decision trees based on what is in scikit-learn.

Typos, etc.:

"a kind of automatic dimension reduction and a kernel-like representation" -- unclear

"domain, So"

**Questions:**

How many trees are used in the random forests of the proposed methods and how are hyperparameters selected for the trees in the random forests?

What is the difference between kernel density decision trees and the work presented in

@inproceedings{Geurts2005,
author = {Pierre Geurts and Lous Wehenkel},
booktitle = {Proceedings of the 22nd International Conference on Machine Learning},
pages = {233-240},
publisher = {ACM},
title = {Closed-form dual perturb and combine for tree-based models},
year = {2005}}

and why is this method of Geurts and Wehenkel not suitable?

---

> ### Author Response · Authors · 2023-11-15
> **Response to review**
>
> In response to weaknesses:
>
> We do not expect consistent advantage for any semi-supervised method. As we stated in the text, the performance of a semi-supervised learning algorithm largely depends on how well its underlying assumptions apply to the data. This is why different methods do well on different data, and why it is common for even well-accepted methods to do worse than purely supervised learning on some data sets. The fact that ours is the best by a notable margin on many data sets is the best we can reasonably expect and is certainly, we believe, enough to demonstrate the value of the method. Our hyperparameter tuning is coarse, but necessary, and we acknowledge this as a limitation. It is also a strength, in some sense, since it allows tuning of the semi-supervised assumptions, which is not possible in methods that lack such hyperparameters.
>
> We will add baselines which are a fully supervised versions of our method, that is, just KDDT trees and forests.
>
> We did not compare to SSL-PCTs because there is no implementation given by the authors, but we will do our best to reproduce it and include it as a baseline.
>
> Showing confidence in the plots in the main paper would make them unreadable. We will include tabular results with standard deviations in the appendix. Between the number of data sets, baselines, and labeled sample counts, there would be a very large number of confidence tests. Do you have a way of presenting this in mind?
>
> We gave transductive evaluation based on the precedent of the closest related method, Levatic et al. (2017), and because it makes it possible to compare to label propagation. However, based on multiple reviewers' requests, we will instead report inductive performance in the revision and, accordingly, leave out label propagation as a baseline.
>
> We are aware and acknowledge that the cross-validation method we use in experiments is not ideal. We only use it for our proposed method, so it is not unfair to the baselines. It is true that parallelization is straightforward, but we are already parallelizing across hundreds of experiment configurations to the limits of the resources we have available, so we cannot afford full cross-validation for the scope of experiments we have shown. In a real application, where there is only one rather than hundreds of cases, so full cross-validation is reasonable.
>
> We will look into these improvements of self-training to include as baselines in the revision.
>
> We will fix the typos etc. Thanks for pointing these out.
>
> In response to questions:
>
> All random forests in this work contain 100 trees. The baseline random forests are fully grown, as is standard for random forests, so there is no hyperparameter selection.
>
> We were not aware of the Geurts2005 paper. Thank you for pointing this out. The formalism for prediction used in that paper is the same as for KDDTs used with a Gaussian kernel. However, KDDTs also use the same kind of deterministic perturbation on the training data during training, which is done using an algorithm proposed by the original KDDT paper. The method of Geurts2005, on the other hand, does normal decision tree fitting. The perturbation aspect of fitting is important to this SSL method because it grows a tree according to the probabilistic interpretation that is then used to assign leaf values. The theory that motivates our leaf assignment strategies would not work without it, so the method of Geurts and Wehenkel is not suitable.

---

> > ### Comment · Reviewer_HSUS · 2023-11-23
> > **Response to authors' comments**
> >
> > Thank you for your replies to my comments. Regarding the issue of confidence intervals: I did not have any particular form of presentation in mind. Using tables seems fine.

---

### Author Response · Authors · 2023-11-15
**Acknowledgment of reviews and plan for revision**

Thanks to all the reviewers for your constructive feedback. We are incorporating as much as we can into a revision that we will upload as soon as it is ready.

We are replying to each of the reviewers to address their comments and start a discussion to gather more feedback.

The biggest change will be adding several experimental baselines requested by reviewers and switching to inductive rather than transductive evaluation of performance, for which we cannot include label propagation as a baseline. Some of the requested baselines do not have public implementations available, and while we will do our best to include these, we cannot guarantee our ability to create a working implementation of all of them. It will also take us some time to implement and test these.

We will also make requested changes to the text, including adding related works, expanding the discussion of them, fixing some typos and unclear mathematical statements, and making some other requested changes.

---

### Author Response · Authors · 2023-11-21
**Gathering feedback for a future submission**

Thanks again to all the reviewers for your comments.

We have decided to target a future venue so that we have enough time to fully incorporate all the feedback from these reviews. We would appreciate any follow-up comments on our revision plans and answers to our clarifying questions before the discussion period closes.